mental health; global mental health; low income countries; ethics; FAIR Guiding Principles

**Author for correspondence:**
Yaara Sadeh,
Email: sadehy@chop.edu

# Opportunities for improving data sharing and FAIR data practices to advance global mental health

Yaara Sadeh[1,2] , Anna Denejkina[3,4,5] , Eirini Karyotaki[6,7] ,
Lonneke I. M. Lenferink[8,9,10] and Nancy Kassam-Adams[1,2,11]

[1]Center for Injury Research and Prevention, Children's Hospital of Philadelphia, Philadelphia, PA, USA; [2]Trauma Data Institute, Lovingston, VA, USA; [3]Graduate Research School, Western Sydney University, Penrith, NSW, Australia; [4]Translational Health Research Institute, Sydney, Australia; [5]Young and Resilient Research Centre, Sydney, Australia; [6]Department of Clinical, Neuro- and Developmental Psychology, Vrije Universiteit Amsterdam, Amsterdam, Netherlands; [7]Amsterdam Public Health Institute, Amsterdam, Netherlands; [8]Department of Psychology, Health & Technology, University of Twente, Enschede, Netherlands; [9]Department of Clinical Psychology, Utrecht University, Utrecht, Netherlands; [10]Department of Clinical Psychology and Experimental Psychopathology, University of Groningen, Groningen, Netherlands and [11]Perelman School of Medicine, University of Pennsylvania, Philadelphia, PA, USA

## Abstract

It is crucial to optimize global mental health research to address the high burden of mental health challenges and mental illness for individuals and societies. Data sharing and reuse have demonstrated value for advancing science and accelerating knowledge development. The FAIR (Findable, Accessible, Interoperable, and Reusable) Guiding Principles for scientific data provide a framework to improve the transparency, efficiency, and impact of research. In this review, we describe ethical and equity considerations in data sharing and reuse, delineate the FAIR principles as they apply to mental health research, and consider the current state of FAIR data practices in global mental health research, identifying challenges and opportunities. We describe noteworthy examples of collaborative efforts, often across disciplinary and national boundaries, to improve Findability and Accessibility of global mental health data, as well as efforts to create integrated data resources and tools that improve Interoperability and Reusability. Based on this review, we suggest a vision for the future of FAIR global mental health research and suggest practical steps for researchers with regard to study planning, data preservation and indexing, machine-actionable metadata, data reuse to advance science and improve equity, metrics and recognition.

## Impact statement

Globally, there is a high burden of mental ill-health, with disproportionate burden in marginalized communities. There is an urgent need to better understand risk and protective factors for mental health and to develop effective strategies to address mental illness, in order to better support individuals, families, and communities. Sharing and reuse of global mental health research data can accelerate collaboration and knowledge development, helping to inform policy decisions, support evidence-based intervention strategies, and allocate resources in an effective and equitable manner to improve mental health outcomes. The value of data sharing and reuse for global mental health is demonstrated by examples of past projects in which data from multiple studies and countries were shared and combined to generate new insights. The FAIR (Findable, Accessible, Interoperable, and Reusable) Guiding Principles for scientific data provide a framework to improve the transparency, efficiency, and impact of research by making data more Findable, Accessible, Interoperable, and Reusable. This review delineates the FAIR principles as they apply to global mental health research, and describes the current state of FAIR data practices in the field, including ethical and social equity considerations in sharing and reusing mental health research data. We describe a number of notable collaborative efforts, often crossing disciplinary and national boundaries, that show the feasibility and promise of improving the Findability and Accessibility of global mental health data, and of building resources and tools that enhance the Interoperability and Reusability of these data. Based on this review we provide a vision for the future of FAIR global mental health research, and suggest practical steps that researchers and research communities can take to improve the FAIR-ness of their data and enhance the impact of their research.

## Introduction

Mental health challenges and mental illness are associated with significant health burden for individuals and societies (Rehm and Shield, 2019; Yang et al., 2021). For those in low- and

middle-income countries (LMIC) and for members of marginalized groups, the burden may be even higher (Ademosu et al., 2021; World Health Organization, 2022). The impact of the global COVID-19 pandemic has exacerbated both overall population-level mental health burden as well as social and economic disparities in populations most affected (Kola et al., 2021; World Health Organization, 2022). The high burden of mental ill-health provides a compelling rationale for optimizing the efficiency and inclusiveness of global mental health research. The WHO's Mental Health Action Plan (World Health Organization, 2013) calls for an enhanced focus on mental health services as well as strengthening information systems, research, and evidence.

Greater sharing of mental health research data can help promote global mental health partnerships and accelerate knowledge development. Preserving and sharing research data and tools makes it easier to assure replicability of findings and to answer key research questions through novel reuse of data. Better understanding of the mechanisms underlying mental health and mental illness allows us to improve diagnosis, interventions, and outcomes (Tenenbaum et al., 2017). Indeed, the value of data reuse to advance mental health research can be seen in prior (usually one-off) efforts that integrated existing individual participant-level data (IPD) across studies and countries to enable new analyses that were not otherwise possible; see Figure 1 for selected examples.

This review focuses on data sharing, preservation, and reuse as a crucial component of optimizing the impact of mental health research globally. We focus on the FAIR Guiding Principles (Wilkinson et al., 2016) for data stewardship because promoting more transparent science by making data more Findable, Accessible, Interoperable, and Reusable (FAIR) can accelerate scientific understanding across mental health research areas. Despite this promise, systematic approaches to data stewardship in mental health research have been sparse to date (Kim and Yoon, 2017). And while funders and journals increasingly call for research data to be shared or archived, this has little value unless accompanied by contextual information (metadata) and tools that make those data interoperable and reusable (Pasquetto et al., 2017).

### Ethical, equity, and social justice considerations

Most mental health research is conducted with human participants who consent to share sensitive information about their lives. Data reuse can be seen as honoring research participants' contributions by maximizing the scientific value of the data they have provided, rather than treating data as an asset "owned" by the researchers who happened to collect it, from which they alone can extract value (Wilbanks and Friend, 2016; Sim et al., 2020). Making data available for reuse can present technical and ethical challenges, such as the extent to which informed consent includes permission for data sharing and reuse (Van den Eynden, 2017) and issues related to data anonymity and deidentification (Curty et al., 2016; Mondschein and Monda, 2019). There are important national and regional differences in legal and regulatory policies in terms of expectations for sharing data as well as restrictions regarding data sharing or reuse. In general, at the time of initial consent, participants should be informed about expectations for sharing or reuse of their data, including measures for anonymizing/deidentifying data (National Health and Medical Research Council, 2018; Rothstein, 2021). Even with consent, researchers must still exercise judgment regarding which data are shared and in what circumstances, to preserve the interests and safety of participants (National Statement on Ethical Conduct in Human Research, 2007).

Data sharing and reuse also require consideration of equity and social justice impact. In terms of equity, LMIC, which currently have the fewest mental health treatment and research options, should be at the forefront of this work and of global partnerships that harness complementary skills and experience of LMIC and high-income country (HIC) partners (Breuer et al., 2019). In data collection, sharing, and reuse, LMIC partners likely bring greater

---

**VALUE OF DATA RE-USE IN GLOBAL MENTAL HEALTH RESEARCH**
**Examples of combining existing individual participant-level data (IPD) from multiple studies to address key questions**

**UNDERSTAND ETIOLOGY, DEVELOPMENT, RISK AND PROTECTIVE FACTORS FOR MENTAL HEALTH**

- IPD from 14 studies in 2 countries: Association between physical & mental health as youth transition into adulthood (Kern et al., 2016);
- IPD from 19 studies in 14 countries: Symptoms & functioning across lifespan in older adults with bipolar disorder (Sajatovic et al., 2019) **
- IPD from 10 studies in 6 countries: Developed prediction tools for PTSD risk after acute injury in adults (Shalev et al., 2019)
- IPD from 18 cohorts in 6 countries: How psychosocial factors (e.g. depression, anxiety) influence cancer risk (van Tuijl et al., 2021)

**ALLOW MORE NUANCED EXPLORATION OF HOW, WHY, AND FOR WHOM MENTAL HEALTH TREATMENTS WORK**

- IPD from 36 trials in 5 countries: Mediators & moderators of treatment effects in evidence-based treatment for PTSD & substance use disorders (Hien et al., 2022)
- IPD from 15 datasets in 10 countries: Use of virtual reality interventions for anxiety disorders (Fernández-Álvarez et al., 2019) **;
- IPD from 11 studies in 10 countries : Impact of psychosocial support interventions for trauma-exposed children in humanitarian settings (Purgato et al., 2018) **
- IPD from 9 trials in 2 countries: Treatment effectiveness, latent trajectories, & predictors of symptom change in youth anxiety (Skriner et al., 2019)

**UNDERSTAND BRAIN STRUCTURE AND FUNCTION RELATED TO MENTAL HEALTH**

- IPD from 45 study cohorts in 14 countries: Identified associations between major depression & structural brain measures (Schmaal et al., 2020) **

**UNDERSTAND DISPARITIES / EVALUATE SOLUTIONS FOR VULNERABLE GROUPS UNDER-REPRESENTED IN RESEARCH**

- IPD from 13 trials in 6 countries: Evaluated potential differential impact based on social disadvantage of parenting program for child conduct problems (Gardner et al., 2019)

**Figure 1.** Value of data reuse in global mental health research. **Includes data from low- to middle-income countries.

expertise than HIC partners regarding contextual influences (i.e., to understand symptoms and expressions of mental illness) and successful provision of high-quality care in scarce-resource settings (Breuer et al., 2019).

To date, marginalized groups have not been well-represented as participants or as investigators in mental health research. An exemplary effort to take into consideration power differentials and historical context is the development of the CARE Principles for Indigenous Data Governance (Research Data Alliance International Indigenous Data Sovereignty Interest Group, 2019). The CARE principles highlight *C*ollective benefit, *A*uthority to control, *R*esponsibility, and *E*thics with regard to information and knowledge that impact Indigenous communities, nations, and individuals. These principles recognize the need to center the rights and interests of Indigenous Peoples in the use of indigenous knowledge and data, and to design data ecosystems that enable this collective benefit and self-determination. The CARE standards are beginning to be operationalized to guide data governance related to health and mental health, for example by tribal governments in the US (Carroll et al., 2022), and participatory planning regarding mental health of Aboriginal and Torres Strait Islanders in Australia (Dudgeon et al., 2021). CARE and FAIR principles are complementary and ideally should be aligned in practice (Carroll et al., 2021).

### The FAIR guiding principles for scientific data

To help frame this review we briefly define key concepts (data and metadata) and then describe the FAIR principles. "Data" can be any representation of information in a formalized manner (Borgman, 2017). In mental health research, data may encompass information collected specifically for the purposes of research (e.g., questionnaire or interview responses, physiological measurements), as well as information gathered from existing sources (e.g., health or administrative records). Raw data are often processed to create new variables for analysis, and data may be captured at varying degrees of granularity (e.g., items vs. scale scores). Both study-level and individual participant-level data (IPD) are important and may be preserved for sharing or reuse (Towse et al., 2021). "Metadata" is data that describes data. Metadata elements describe data's provenance by capturing study-level characteristics (i.e., information about study design or social context), as well as specific variables (measures, items) and how these were collected or derived. Metadata should provide rich descriptive information that helps researchers find, understand, and reuse the data. Ideally, metadata are machine-readable and also able to be adapted/displayed for meaningful human use (Arslan, 2018).

The FAIR Guiding Principles (Wilkinson et al., 2016) were created by a diverse group of stakeholders (including researchers, publishers, and funders) to provide guidance for the management and stewardship of scientific data. Because they are intended to apply across scientific disciplines and across many types of data and research, Findability, Accessibility, Interoperability, and ReUsability (F, A, I, and R) represent foundational principles rather than specific, prescriptive rules. The FAIR principles put particular emphasis on using automated, machine-actionable processes that allow researchers to preserve, find, and use existing data resources (Wilkinson et al., 2016). Table 1 presents each principle with a brief definition and examples illustrating its application in mental health research. For a comprehensive description of the FAIR principles, see Wilkinson et al. (2016) and the European Commission, Directorate-General for Research and Innovation report (2018). It is

**Table 1.** FAIR data principles with definitions and application to mental health research

| Principle | Brief definition | Examples of application to mental health research |
|---|---|---|
| F: Findable | Researchers who may want to use the data are able to discover that the data exist. Machine-readable metadata are key to Findability | ■ Broad data repositories may be searchable by mental health topics, for example, Inter-University Consortium for Political and Social Research (ICPSR) (https://www.icpsr.umich.edu/web/ICPSR/search/studies?q=mental+health) <br> ■ Some indexes have been created for specific topics, for example, for UK longitudinal studies, the Catalog of Mental Health Measures (https://www.cataloguementalhealth.ac.uk/), and the Global Collaboration on Traumatic Stress' index of data resources (https://www.global-psychotrauma.net/data-sets) |
| A: Accessible | There is a clear means of gaining or requesting access to the data | ■ Some data are openly available, for example, within ICPSR (see above) one can filter for "public use" <br> ■ Some data resources provide clear procedures for requesting access, for example, Netherlands Study of Depression and Anxiety (https://www.nesda.nl/nesda/wp-content/uploads/2019/07/NESDA_policy_-data_access.pdf); Child Trauma Data Archives (https://childtraumadata.org/use-pactr-data) |
| I: Interoperable | Data are stored and described (metadata) with human- and machine-readable standards, usable across a variety of software systems | ■ National and institutional data repositories that include mental health data generally maintain data usable across a range of present and future software systems <br> ■ Machine-readable metadata standards in the social sciences (e.g., Data Documentation Initiative https://ddialliance.org/) may be applicable to mental health research |
| R: Reusable | Data are organized and described (metadata) with sufficient detail for actual use to replicate findings or address new research questions | ■ No metadata standard specifically designed for mental health research, but there are nascent efforts to create standards for psychological research datasets (e.g., https://psych-ds.github.io/) <br> ■ Projects in specific mental health research areas are developing metadata and methods to support data reuse, for example, in child traumatic stress (Kassam-Adams et al., 2020) or substance use disorders (Susukida et al., 2021) |

crucial to understand that FAIR is not the same as "open"; data can be FAIR but not open (and vice versa). While related to the broader movement for more open and transparent science, none of the principles require data to be openly or freely available. Rather, they emphasize the need for clarity and transparency regarding access and reuse conditions. Many mental health data resources cannot be fully open, but nearly all can be made accessible. Transparent but controlled access to data may allow participation from a wide range of sectors in society (Mons et al., 2017).

This review provides an overview of the current state of the FAIR principles as applied in global mental health research. We explore current practices amongst mental health researchers and research communities and provide examples of notable efforts to create FAIR data resources. Finally, we outline a vision for next steps to make mental health research practices more FAIR.

## How "FAIR" is global mental health research?

### Current state of the FAIR principles in global mental health research

Are mental health data resources findable? Unfortunately, there are few resources that help investigators find applicable and potentially reusable mental health data. As noted in Table 1, broad (national or institutional) repositories may include mental health data that are findable via searching within those systems, and a few specific indexes have been created.

Accessibility is better managed, at least with regard to mental health data within well-established repositories. Most repositories allow data contributors to specify how data should be disseminated, including registered or restricted access to mental health research datasets. And most have clear processes for data access, ranging from formal data request and approval processes to publicly accessible data that can be directly downloaded (see Table 1 for examples). On the other hand, when data are informally "available upon request" by authors or investigators, there are widely varying practices and often little information as to how the data may be accessed or conditions for use.

Interoperability is generally assured for mental health data maintained in well-established repositories, but when individual researchers hold their own datasets for informal sharing, they are unlikely to have the resources to maintain and update multiple (changing) data formats across time. Interoperability of metadata depends on common, machine-readable metadata standards. Metadata standards for the social sciences (e.g., Data Documentation Initiative, https://ddialliance.org/) exist but are not yet optimized for mental health research.

The reusability of mental health data is most severely impeded by the lack of clear documentation. Documentation (metadata) is often collected in codebooks or data dictionaries, but in current practice, these are generally unstandardized, not machine-readable, and lacking essential information to make sense of the data (Arslan, 2018; Towse et al., 2021). Across the FAIR principles, the absence of commonly agreed standards for mental health research metadata (about studies and about variables) hampers findability, interoperability, and (most notably) efficient and effective data reuse. A few efforts that bring together individual participant-level data across studies, for example, in traumatic stress (Kassam-Adams et al., 2020) and in substance use disorders (Susukida et al., 2021), have begun to develop ways to describe common concepts with metadata, but there is still a lack of collectively agreed terminologies, guidelines, and protocols (Fortier et al., 2017).

### Awareness, support, and practice amongst mental health research stakeholders

The discipline of psychology has been a leading force in the movement toward data sharing, particularly with regard to replicability of scientific findings (Nosek et al., 2022), and "open science" technologies closely related to FAIR practices (Christensen et al., 2019). (A full description of open science initiatives in mental health is beyond the scope of this paper – for expanded information see: Kathawalla et al., 2021; Robson et al., 2021.) Yet perceptions and expectations regarding data sharing and reuse vary widely in mental health research communities. In a 2021 survey of psychology researchers from 31 countries, the majority indicated that they had archived, deposited, or published a dataset for others to access but noted a lack of standardization of practices within their research group (Borghi and Van Gulick, 2021). Surveys have found barriers to data sharing including uncertainty about implementing new research practices to support sharing, beliefs that these practices are unnecessary or burdensome (Washburn et al., 2018), concerns about others stealing their ideas (Houtkoop et al., 2018), and ethical/legal concerns (Tedersoo et al., 2021). A study examining the quality of shared data in psychology found that 51% of datasets were incomplete and 68% had limited reusability, that is, proprietary software, nonmachine-readable data, metadata not sufficiently informative to understand the dataset (Towse et al., 2021).

Beyond the practices of individual researchers, a variety of national policies impact mental health data sharing and reuse (Packer, 2010; Fernando et al., 2019), including regulations on research ethics, human subjects protection, and data privacy (e.g., GDPR in the EU, HIPAA for health data in the US). The WHO's (2022) policy and implementation guidance states that data sharing is an obligation for WHO staff and researchers funded by WHO. Other large nongovernmental stakeholders have promoted data sharing; for example, the Scientific Electronic Library Online (SCIELO; Packer, 2010) initiative in 17 countries (primarily Latin America) now supports a data repository. Some national and nongovernmental research funders require data sharing or formal data management plans as a condition of funding, for example, *North America* – National Science Foundation, 2011; National Institute of Health, 2022; *South America* – The United Nations Economic Commission for Latin America and the Caribbean, 2022; *Africa* – see review by Obiora et al., 2021; *Europe*: European Research Council, 2022; *Australia* – National Health and Medical Research Council, 2019. Even when not mandated, investigators who systematically archive and share data may become better candidates for funding (Bosma and Granger, 2022).

A growing number of journals relevant to mental health research have data archiving policies (Cooper and VandenBos, 2013; Nuijten et al., 2017; Hardwicke et al., 2018). International scientific societies are also active in promoting open and collaborative science and data sharing, for example, initiatives by the Society for the Improvement of Psychological Science (SIPS: https://improvingpsych.org), and the Global Collaboration on Traumatic Stress (https://www.global-psychotrauma.net/fair).

### Current notable efforts that advance FAIR data practices in mental health research

Despite the challenges described above, there are notable efforts to create resources that preserve, describe, share, and/or support reuse of mental health data. To demonstrate the feasibility and promise of this work, we highlight some noteworthy and relevant examples

here (This is not an exhaustive list.) Most of these efforts involve substantial collaboration across disciplines and national borders; they vary in their scope of coverage, ease of access, and cost. While many are not explicitly defined as "FAIR" initiatives, each of the efforts described here provides tools or data resources that put the FAIR principles into practice.

## Findable and Accessible

Findability is a crucial first step (Tenenbaum et al., 2017) and global mental health data can be made more findable and accessible in a variety of ways. We focus here on efforts that have mapped specific mental health topic areas, generally by indexing and linking to data sources, allowing researchers to discover, request, or reuse data.

- The Programme for Improving Mental Health Care (PRIME) is an LMIC-led partnership that provides research evidence regarding mental healthcare in Ethiopia, India, Nepal, South Africa, and Uganda. The PRIME partnership proactively addressed the issue of data ownership with a clear policy (http://bit.ly/2BwiZu2) on data sharing and publication. For non-PRIME parties, data from PRIME is available for future use upon request (Breuer et al., 2019).
- The Human Heredity and Health in Africa (H3Africa) Initiative (https://catalog.h3africa.org/) is facilitating the study of genomics and environmental determinants of common diseases with the goal of improving the health of African populations. Data and biospecimens from H3Africa projects, including several related to mental health, are available for further use, with access controlled by a Data and Biospecimen Access Committee.
- The Catalog of Mental Health Measures (https://www.cataloguementalhealth.ac.uk/) describes mental health and well-being measures in British cohort and longitudinal studies. It presents detailed information about these measures and the studies in which they were used, how to access or request data from each study, plus training and resources on conducting longitudinal mental health research.
- The Global Collaboration on Traumatic Stress FAIR Data Theme provides a growing index of data resources (single studies and collections of datasets) that may be useful for secondary analysis (https://www.global-psychotrauma.net/fair-data-sets). The index describes each data resource and how it can be accessed.
- Kievit et al. (2022) provide an index of large developmental data sets that include adolescents and can be used for secondary analysis, with information on data access.
- The Single Case Archive (www.singlecasearchive.com) assembles published psychotherapy case studies. Researchers can use systematized, searchable case-descriptive information to aggregate findings across sets of cases (Desmet et al., 2013).

## Interoperable and ReUsable

We highlight efforts to create integrated data resources with the explicit intention of supporting ongoing access, reuse, and interoperability. This includes resources that pull together existing study- or participant-level data for reuse, as well as projects that explicitly aim to generate data to be shared with the field.

(a) Resources that collect study-level aggregate findings for reuse:

- The PTSD Trials Standardized Data Repository (PTSD-Repository: https://ptsd-va.data.socrata.com/) includes study-level data from almost 400 randomized controlled trials (RCTs) of interventions for PTSD in adults. Users can view, download, and manipulate repository data for a variety of purposes, including practitioners designing treatment plans for patients and investigators conducting exploratory analyses to identify common variables and inform future trial design (O'Neil et al., 2020).
- The Maelstrom Research catalog (www.maelstrom-research.org) indexes epidemiological data from population-based cohort studies, including several relevant to mental health. It facilitates the exploration of harmonization potential across cohorts, subpopulations, and data collection events, and offers open-source software for researchers to develop their own catalogs and metadata fields (Bergeron et al., 2018).

(b) Resources that bring together individual participant data (IPD) to support harmonization and novel analyses:

- The Psychiatric Genomics Consortium (PGC; https://www.med.unc.edu/pgc/pgc-workgroups/) is the largest consortium in the history of psychiatry, with >800 investigators from 38 countries. PGC members share raw genotype data processed using a uniform quality control and analysis pipeline (Sullivan et al., 2018). The PGC supports meta- and mega-analyses of genomic data, with workgroups for specific disorders including autism, attention-deficit hyperactivity disorder, bipolar disorder, major depressive disorder, PTSD, and schizophrenia (Logue et al., 2015; Sullivan et al., 2018).
- The MetaPSY database (https://evidencebasedpsychotherapies.shinyapps.io/metapsy/) includes IPD from 411 RCTs of psychological treatment for depression and suicide prevention in adults (Cuijpers et al., 2020), with plans to expand to other disorders (e.g., anxiety, PTSD, insomnia, grief). Its interactive website (Cuijpers et al., 2019, 2020) uses models that combine available predictors, moderators, and interactions to generate outcome predictions, and allows users to perform novel meta-analyses. The database has been used to answer a wide range of research questions regarding target groups, settings, depression subtypes, and therapy characteristics to help determine which psychotherapies are most effective for whom (Cuijpers, 2017).
- The database of RCTs of psychosocial interventions for suicidal thoughts and behavior provides an ongoing research resource that can be accessed via request to facilitate systematic reviews and meta-analyses and stimulate research in suicide prevention (Christensen et al., 2014).
- The Global Collaboration on Traumatic Stress supports a growing collection of integrative IPD data projects, currently in the areas of child trauma, traumatic grief, adult trauma interventions, and veteran mental health (https://www.global-psychotrauma.net/fair). The Child Trauma Data Archives project (https://childtraumadata.org/) includes IPD from >30 prospective studies in five countries plus a new archive of child trauma intervention studies. The Measurements Archive of Reactions to Bereavement from Longitudinal European Studies (MARBLES: https://www.uu.nl/en/research/the-marbles-project) is pooling data from observational studies on grief, and creating another archive pooling IPD from grief treatment studies (https://people.utwente.nl/l.i.m.lenferink?tab=projects). The

global research consortium for Treating and Understanding Trauma Treatment Interventions (TUTTI: https://www.tuttirc.com/) brings together IPD from RCTs of adult trauma treatment (Wright et al., 2022) to enable new analyses of clinically relevant moderators of treatment effects. The International Veteran Dataset Initiative (IVDI) is creating an international dataset to facilitate integrated analyses regarding military mental health.

(c) Projects that collect mental health data with sharing and reuse as an explicit project aim:

- The Global Psychotrauma Screen (GPS) was developed by the Global Collaboration on Traumatic Stress. Its adult version (https://www.global-psychotrauma.net/gps), available in more than 20 languages, is a simple, cross-culturally valid, easy-to-administer tool that screens for a wide range of potential outcomes of trauma (Olff et al., 2020). GPS datasets with more than 7,000 adult participants worldwide are openly available for reuse by investigators (https://osf.io/untsy/files/osfstorage).

- The Advancing Understanding of RecOvery afteR traumA (AURORA) project is a US emergency department-based study (target $N = 5,000$) collecting genomic, neuroimaging, psychophysical, physiological, neurocognitive, and self-report data over 1-year post-trauma. The multilayered AURORA dataset is designed for use by the scientific community to study posttraumatic neuropsychiatric sequelae (McLean et al., 2020).

## Conclusions and next steps: Moving toward more FAIR (and equitable) data practices in global mental health

Embracing FAIR principles by preserving, sharing, and reusing mental health data is essential to the short- and long-term impact of our scientific work. We have numerous examples demonstrating the value of integrating existing research data for novel analyses that enhance our understanding of etiology, risk, and protective factors for mental health, and that advance clinical practice. Growing calls from key research funders for data sharing and accessibility highlight the need for every research team to incorporate more FAIR data practices.

This review points to both challenges and opportunities for implementing FAIR data practices in global mental health research. Our field has not yet developed a common expectation and culture of FAIR data practices, nor the widely available tools and resources that would allow every mental health researcher to easily engage in FAIR practices. Specific challenges and gaps include varying support for data sharing and reuse, not planning for preservation or reuse when designing or conducting research, lack of standard practices for data management and preservation even amongst research teams and communities, poor findability of global mental health data resources due to a lack of common standards for describing our studies and our data points, and varying regulatory standards for sharing mental health research data. Yet there are a range of (often topic-specific) projects that are already providing useful resources, and that constitute a strong proof-of-concept for both the feasibility and the value of broader adoption and use of FAIR data practices across the field of global mental health. Many of these international projects and data resources are led by institutions and researchers within HICs. The field would be strengthened by having more global mental health research partnerships that are led by, and harness the unique expertise and knowledge of, researchers within LMICs and other marginalized communities.

Based on this review we have identified opportunities for researchers to move toward more FAIR data practices. Table 2 presents a vision for FAIR global mental health data with suggestions for practical next steps by researchers. The vision

**Table 2.** FAIR global mental health data: Vision for the future and practical next steps

|  | Vision for the future: What would it look like if global mental health data were FAIR? | Practical next steps: What can researchers do now? |
|---|---|---|
| Study planning | • Mental health research studies – of any size – are planned with data preservation, sharing, and reuse in mind | • Educate yourself and your team about FAIR data practices<br>• When initiating collaborative work, consider data sharing and equity especially amongst collaborators of varying access and resources, for example, Kumar et al. (2022)<br>• Examine your own data practices across the research lifecycle: Where can you be more FAIR? Consider benefits, as well as challenges and how you can address them<br>• Start with the basics: well-organized data understandable for humans and machine (Broman and Woo, 2018)<br>• Learn more about applying FAIR principles here: https://www.howtofair.dk/what-is-fair/ |
| Data foster preservation and indexing | • Global mental health research datasets are collected in well-curated repositories that provide long-term preservation and machine-readable persistent identifiers<br>• Accessible mental health data resources are indexed within and across specialty areas to increase findability | • Make a map of your specialty area: What datasets are accessible? Where are the gaps?<br>• Collaborate with colleagues to create an accessible, updateable, online index of available data resources in your topic area and where they can be found<br>• Deposit your data in a reliable repository at your institution or elsewhere – look for the CORE Trust Seal for repositories<br>• Choose a repository that can issue a DOI so your data can be easily cited and their impact tracked. Learn more: https://www.dcc.ac.uk/guidance/how-guides/cite-datasets<br>• If sharing individual participant-level data is not feasible for ethical or legal reasons, share/deposit aggregated study-level data and metadata, for example, O'Neil et al. (2020) |

*(Continued)*

**Table 2.** (*Continued*)

| | Vision for the future: What would it look like if global mental health data were FAIR? | Practical next steps: What can researchers do now? |
|---|---|---|
| Machine-actionable metadata | • Researchers consistently use a common set of machine-actionable metadata standards for global mental health research data that ensure findability and reusability across repositories/data resources | • Facilitate future use of your datasets (by your own team and others) by describing your study and data with high-quality metadata<br>• Document data with machine-readable metadata about their context, provenance, and details. Learn more about documentation and metadata here: https://rdmkit.elixir-europe.org/metadata_management |
| Data reuse to advance science and improve equity | • Reuse of global mental health research data is valued and results in novel findings not possible from single studies<br>• Data preservation, sharing, and reuse takes equity into account and includes active participation and voice of marginalized groups | • Consider using existing data to address key research questions and/or to refine your design of new research<br>• In your role as journal editor or reviewer, promote standard machine-actionable citation of datasets to track impact and promote data reuse. Learn more here: https://force11.org/info/joint-declaration-of-data-citation-principles-final/<br>• Build and support partnerships led by researchers from lower resource settings, for example, Breuer et al. (2019) |
| Metrics and recognition | • Producing useful data resources is a key career milestone for global mental health researchers<br>• There are well-accepted metrics for the scientific and societal impact of these resources | • In your role as mentor, supervisor, and peer reviewer, thinks beyond publication when considering colleagues' scholarly impact. Support the idea that creating reusable data resources is a marker of productivity and impact |

*Source:* Adapted and expanded from Kassam-Adams and Olff (2020).

is adapted from a framework for FAIR traumatic stress research (Kassam-Adams and Olff, 2020), broadened to address the field of mental health, with expanded information and links to support specific actions. The framework addresses five key themes: Study planning, data preservation and indexing, machine-actionable metadata, data reuse of advance science and improve equity, and metrics and recognition within academia. Across these themes, Table 2 lists feasible next steps that focus on educating oneself and one's research team; building collaborative projects that capitalize on expertise within specialty areas in global mental health; considering data sharing, reuse, and equity in planning for research and collaboration across the data lifecycle; and individual actions (in our roles as investigators, mentors, peer reviewers, editors) that can help shift the culture and practice of our field.

In conclusion, building the FAIR principles into the way we create collaborations, design and conduct studies, manage data, disseminate findings, and measure the academic and societal impact of global mental health research will require a continuing culture shift as well as systemic changes by larger stakeholders (academia, journals, funders). It will also require collective and collaborative action by researchers, research teams, and scientific societies. The choices we make are important, as they will impact the pace of future research advances that allow us to effectively address the huge mental health burden borne by so many in our society. In this review, we have provided examples of promising collaborative efforts upon which we can build, and exemplars within subfields of global mental health research that can inspire work in other topic areas and research communities. FAIR-ness is a continuum (i.e., not "all-or-nothing"); thus our goal should be to increase the FAIR-ness of our practices, considering the larger context and specific challenges. We hope that readers will be inspired to join the efforts described here and to adapt these approaches to address gaps in FAIR data practices in their own area of mental health research.

**Open peer review.** To view the open peer review materials for this article, please visit http://doi.org/10.1017/gmh.2023.7.

**Acknowledgements.** We gratefully acknowledge the ongoing efforts of the Global Collaboration on Traumatic Stress FAIR Data Theme Workgroup which have informed the views and analysis reported here.

**Author contributions.** All authors contributed substantially to the conception or design of the work. Y.S. and N.K.-A. drafted the work. A.D., E.K., and L.I.M.L. revised it critically for important intellectual content. All authors had final approval of the version to be published and agreed to be accountable for all aspects of the work.

**Financial support.** This research received no specific grant from any funding agency, commercial, or not-for-profit sectors.

**Competing interest.** The authors declare none.

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
