## [Reviewer Report]

*Comments to Author*: The authors’ stated aim of the manuscript is to review “data sharing, preservation, and re-use as a crucial component of optimizing the impact of mental health research globally”. The Findable, Accessible, Interoperable, and Reusable (FAIR) principles are introduced. Sharing data in ways that are compatible with the FAIR principles is intended to increase efficiency and inclusiveness of research by improving replicability of studies, encouraging novel re-use of data, and enabling cross-disciplinary studies. The intended result is to improve diagnosis, interventions and outcomes. The main challenges/barriers to using the FAIR principles are ethical considerations including consent for resharing. The primary ethical arguments for using FAIR principles are potential contributions to equity and social impact.

In the second section the authors further discuss concepts related to FAIR principles and their compatibility with mental health research. FAIR represents principles rather than rules or guidance for implementation of open data in a specific context. Furthermore, the authors emphasize that FAIR requires clarity and transparency, but does not require that all data be open which is an important consideration for mental health research, as they state “Many mental health data resources cannot be fully open but nearly all can be made accessible”. A third section outlines initiatives, institutions, and key literature related to mental health research related to each of the FAIR elements. Table 1 appears to organize the observations of the authors on the current status of open access to mental health data organized by each element of FAIR. Proposed next steps are presented in table 2 organized by a framework adapted from the literature that identifies FAIR-relevant research entry points. The authors conclude that the review has “provided examples - across the field of mental health research – of collaborative efforts to develop tools and resources that put the FAIR data principles into action”.

Addressing the timely and important issue of improving research with open data access, the authors make a novel contribution by investigating the current status of mental health data resources using widely recognized FAIR principles. They raise questions that are highly relevant to the field and that are of high concern to institutions that fund mental health research among other audiences.

At first reading of the manuscript, it appeared that there is a missing critical discussion section that would be expected in an article of this type. I was surprised to find a good deal of discussion and the major contribution of the article somewhat hidden away in Tables 1 & 2. In revising the paper, the authors should focus on fully developing the discussion on current status and field-specific challenges in the body of the text. Then, the focus should be on the clear and logical links of the analysis to the novel proposed implementation guidance framework now contained in Table 2. The audience would be specifically interested in how each of these authors’ work;“Kassam-Adams & Olff 2020; 1Broman& Woo, 2018; 2Towse, Ellis & Towse, 2021;3O’Neil et al., 2020.”; was adapted and why. This proposed framework should be clearly identified as the “Opportunities” promised in the paper’s title.

Review Question 1:

For global reviews, how well does the review cover global content in the inclusion of research, presentation of results, and/or in the discussion and implications? And how could this be improved/expanded?

The scope and implications of the review is relevant to a global audience. Global examples in the current manuscript are limited and suggest the authors redouble efforts to include any others that may be currently missing. The authors discuss the survey conducted by Borghi & Van Gulick, 2021 that found “31 countries indicated that they archived, deposited, or published a dataset for others to access but noted a lack of standardization of practices within their research group.”. One global project is mentioned – the Global Collaboration on Traumatic Stress’ FAIR Data workgroup (https://www.global-psychotrauma.net/fair). One bullet point in table 2 states “Global mental health research datasets are collected in well-curated repositories that provide long-term preservation and machine-readable persistent identifiers”.

In the revision, the authors may want to expand on analysis of global data access in comparison with promising discipline or community specific data sharing and collaboration practices. For example, the authors mention national policy frameworks and national research funding agencies, but additional analysis would be welcomed describing what is global and what is particular to different nations. Similarly, the authors state “The CARE Data Principles for Indigenous Data Governance exemplify these aspects of data stewardship (Carroll et al., 2020)”, but I believe additional analysis of how, why or what these community standards contribute globally would strengthen the paper for this journal.

Additional suggested considerations in revision:

In the Introduction, suggest refined focus on the most important review questions on what are fundamental and specific problems for mental health research. The contribution of the paper bridges a perceived gap between principles and practice that might be captured in more questions addressing the demand for guidance and rules applicable to mental health research and how mental health data, “can be made accessible”. The issue of obtaining consent for resharing seems of such fundamental importance, that one might expect much more of a focus on this in critical discussion. A greater focus on what are the important questions would help guide the reader. The value of reusing data is discussed at length in introduction with a focus on “individual participant-level data (IPD) from multiple studies”. This level of detail seems more appropriate in the current section 3.

Additional attention to Defining Key concepts in section 2 is suggested. Table 1 column 3 does not always appear as clear examples related to column 2. For example, metadata is discussed in “R” but is part of the definition of “I”. If the table intends to give definitions in column 2 and then clear examples in column 3, then it is suggested to choose the most illustrative examples with precise language to articulate what the principle might mean in mental health practice. Perhaps taking this information out of the table and focusing on a limited number of key concepts and there relation to mental health research would strengthen this section.

Section 3.1 does seem to need a table or more detail on the comprehensiveness of the review. Are the initiatives and organizations mentioned examples or do they represent the totality of work relevant to mental health research currently? For example, the authors cite Christensen et al., 2019 “Several organizations have emerged to provide training and support for open science technologies”, and it may be helpful to readers to present these in a table or other reference. Their relation and relative value with respect to the Open Science Framework would be of interest and potentially important to the manuscript’s argument for mental health research opportunities organized in a framework. Why is there no mention of GitHub or other open science platforms used by diverse researchers? If these are not a comprehensive survey, but only ‘noteworthy examples’ – this should be more explicit in the text.

Further analysis of the completeness and comparison of examples in section 3.2 are welcome. This might be accomplished by organizing and expanding on some of the content in Table 2 but organized by the FAIR principles in table 1. Consider the organization of this section as leading the reader to the set of categories proposed in column one of table 2. The “current state” now reads somewhat as a list. Concepts like ‘Meta data’ appear in different rows but there is not enough detail to clearly understand why the concept spans these rows. Other authors evaluating FAIR practices within their disciplines often contrast FAIR with another discipline specific framework or a relevant set of criteria/categories. What do the authors suggest as a guiding framework for mental health research that could be contrasted against FAIR?

Section 4 leaves some important questions raised in the Introduction unanswered or left in the tables. It is missing a critical discussion of the gaps and dominant or weaker approaches to address the identified challenges and barriers. For example, more discussion ideas presented in the background such as data“sharing and re-use “baked in” from the beginning as an explicit project aim” would be welcome.

The Conclusion currently states that the paper “provided examples - across the field of mental health research – of collaborative efforts to develop tools and resources that put the FAIR data principles into action”. With revision, the paper can do this well and more. Consider ‘Next Steps’ to be included in conclusions to strengthen what are now generalities.

Overall, this very interesting paper deserves great attention to revisions. The ‘opportunities’ may be strong enough to be the foundation of a very useful framework for guiding the implementation of open data access in mental health research. Perhaps the authors should consider this aim as there is need for a -- discipline specific framework or a relevant set of criteria/categories – to contrast against FAIR to strengthen the analysis – and this adapted frame could be introduced earlier in the paper.

Best wishes to the authors as they consider the revisions and look forward to reading the next version of the paper.

---

## [Reviewer Report]

*Comments to Author*: Dear Authors, please follow the suggestions of Reviewer #1 answering his/her questions.

---

## [Reviewer Report]

*Comments to Author*: I am convinced of the importance of this review for a broad audience from those interested in more efficient use of research resources to those interested in innovative approaches. Readers will find the tables and examples useful.

On page 8, I would consider removing "current state" and instead "overview of FAIR principles as applied...".

A question remains about how the notable examples beginning on page 11 were identified. Are they all the authors could find? Are they everything the authors have seen in professional gatherings? How would one go about making a more exhaustive list and perhaps who should be responsible for better accounting of available repositories?

FAIR principles hints at the social justice concerns regarding best use of data, and I think think the article does well discussing importance, challenges and potential of open science trends for global mental health.

---

## [Reviewer Report]

*Comments to Author*: Dear authors:

Your current manuscript version requires minor revision requested by the reviewer.

There are other important recommendations:

1. Please, after the introduction could you clarify some methodological details of your review? For example, a) methods for finding and reviewing the literature and b) the type of literature review according to: Grant, M.J. and Booth, A. (2009), A typology of reviews: an analysis of 14 review types and associated methodologies. Health Information & Libraries Journal, 26: 91-108. https://doi.org/10.1111/j.1471-1842.2009.00848.x

2. In the point 1.1 Ethical, equity, and social justice considerations, your examples are addressing uniquely the English-speaking middle- and low-income countries. Could you detail examples from non-English-speaking middle- and low-income countries?

3. In the point 2.3 Current notable efforts that advance FAIR data practices in mental health research; could you detail examples from non-English-speaking middle- and low-income countries?

Thank you very much.